# Human Nutrition Research in the Data Era: Results of 11 Reports on the Effects of a Multiple-Micronutrient-Intervention Study

**DOI:** 10.3390/nu16020188

**Published:** 2024-01-05

**Authors:** Jim Kaput, Jacqueline Pontes Monteiro

**Affiliations:** 1Vydiant Inc., Dallas, TX 75201, USA; 2Faculty of Medicine of Ribeirão Preto, Department of Pediatrics, University of São Paulo, Ribeirão Preto 14049-900, SP, Brazil; jacque160165@gmail.com

**Keywords:** personalized nutrition, nutrigenomics, n-of-1 research

## Abstract

Large datasets have been used in molecular and genetic research for decades, but only a few studies have included nutrition and lifestyle factors. Our team conducted an n-of-1 intervention with 12 vitamins and five minerals in 9- to 13-year-old Brazilian children and teens with poor healthy-eating indices. A unique feature of the experimental design was the inclusion of a replication arm. Twenty-six types of data were acquired including clinical measures, whole-genome mapping, whole-exome sequencing, and proteomic and a variety of metabolomic measurements over two years. A goal of this study was to use these diverse data sets to discover previously undetected physiological effects associated with a poor diet that include a more complete micronutrient composition. We summarize the key findings of 11 reports from this study that (i) found that LDL and total cholesterol and fasting glucose decreased in the population after the intervention but with inter-individual variation; (ii) associated a polygenic risk score that predicted baseline vitamin B12 levels; (iii) identified metabotypes linking diet intake, genetic makeup, and metabolic physiology; (iv) found multiple biomarkers for nutrient and food groups; and (v) discovered metabolites and proteins that are associated with DNA damage. This summary also highlights the limitations and lessons in analyzing diverse omic data.

## 1. Introduction

Vitamins and minerals have been studied for over 100 years; yet, much remains unknown about their roles in health and diseases. Large percentages of the American population report inadequate intake (defined as less than the estimated average requirement (EAR)) of specific micronutrients related to immune function [1] and to other chronic and polymorbid diseases [2]. Adapting von Liebig’s law of the minimum [3] from plants to humans, the addition of single or a few micronutrients at recommended daily intakes may expose the physiological effects caused by another limiting micronutrient in the diet. In in silico analysis of protein/gene/pathway/disease, databases support the concept of system over reductionist biochemical reactions. Almost 4000 proteins bind with one or more of 49 vitamins, minerals, and in vivo synthesized cofactors [4]. These co-factor binding proteins contribute to ~1250 diseases, which was determined through mapping their genes to disease loci and linking metabolic processes involved in disease using network and pathway analyses [4].

In *The Structure of Scientific Revolutions* [5], Kuhn described how “normal science” was interrupted by periods of “revolutionary science” resulting in new paradigms and ways of conducting research. Notwithstanding the extensive cofactor/gene/pathway literature base, most nutritional-intervention studies add one or a few micronutrients to an existing diet and analyze the population-level responses of specific biomarkers or pathways. While hypothesis-driven research has been a foundation of biomedical research, genomics, proteomic, metabolic, and other omic technologies are revolutionizing biomedical studies. With some notable exceptions (e.g., [6,7]), nutrition research has been slow in adopting big data strategies and systems approaches [8]. Our teams conducted a broad, data-driven program of micronutrients (Appendix A).

The specific goal of the multiple-micronutrient intervention summarized here was to use diverse data sets to discover previously undetected physiological effects associated with a poor diet supplemented with a more complete micronutrient composition. The nutritional-intervention study in children and teens analyzed the effects of 12 vitamins and five minerals on physiological systems (Box 1). The n-of-1 design accounts for inter-individual variability because each participant is his/her own control. Population-level data were also analyzed by aggregating data from all participants before and after the intervention, and after a washout period. Twenty-six types of data including clinical measures, whole-genome mapping, whole-exome sequencing, and proteomic and a variety of metabolomic measurements were collected in the first year and in the replication arm of the study (Table 1).

In this article, we summarize the key findings of 11 reports from this study (Appendix A) that (i) found that LDL and total cholesterol and fasting glucose decreased in the population after the intervention, but with inter-individual variation; (ii) identified and associated a polygenic risk score that predicted baseline vitamin B12 levels; (iii) discovered metabotypes linking diet intake, genetic makeup, and metabolic physiology; (iv) found multiple biomarkers for nutrients and food groups; and (v) discovered metabolites and proteins that are associated with DNA damage. This summary also highlights the limitations and lessons in analyzing diverse omic data.

Box 1Experimental design.

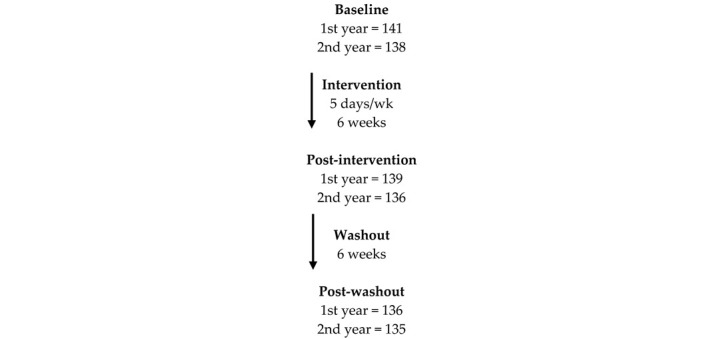



## 2. Rationale and General Experimental Design

### 2.1. Rationale for the Intervention

An n-of-1 experimental design was used without a control arm in each of the two successive years during the same season providing a replication arm. Children and adolescents were recruited from three schools in the west side of Ribeirão Preto, São Paulo, Brazil. Volunteers were of 9–13 years, 11 months, and 29 days old at study initiation, and they were clinically stable. The selection of the age group was based on the high prevalence of being overweight and obese, which is consistent with trends in the child and general Brazilian population and on the poor quality of their diets ([9]—accessed on 15 November 2017)]. In addition, this age group is under-represented in nutrition studies. The project was conducted using principles of community-based participatory research [10,11], that included planning sessions with principals, teachers, and parents at the three schools along with post-experiment health fairs to provide results to each interested parent/participant.

The intervention was a tablet which consisted of 12 vitamins and five minerals (Nestrovit^TM^, Vers-chez-les-Blanc, Switzerland) provided 5 days per week for 6 weeks, followed by a 6-week washout for participants of 9 to 13 years (Box 1). The vitamins in the supplement were retinol palmitate, α-tocopherol acetate, tetrahydrofolic acid, thiamine pyrophosphate, flavin adenine dinucleotide, nicotinamide diphosphate, pyridoxal 5-phosphate, methylcobalamin, cholecalciferol, biotin, pantothenic acid, calcium ascorbate, phosphorus, iron, magnesium, and zinc (see Appendix A). The micronutrient-based supplement was chosen based on the micronutrients’ role of binding proteins and their co-factors that affect multiple pathways, networks, and diseases [4]. In addition, Nestrovit^TM^ was accepted by children and adolescents further enabling compliance. Supplementation at essentially 100% of the daily recommended dose (but less than upper tolerable level for most nutrients) for 6 weeks is a type of dietary challenge, which are typically short term (e.g., a meal) [12,13]. Graduate students associated with the project assured that the students ate the tablets with 98% compliance. The students attended school continuously for the 12-week period.

Assessments at baseline, after 6 weeks of intervention (with supplement) and after 6 weeks of washout (no supplement), in two consecutive years were conducted at the clinic and included food intake through a revised Brazilian healthy eating index, food frequency questionnaire (FFQ), 24 h recall, socioeconomic status, anthropometric analysis, blood sampling for complete blood counts and standard clinical blood analysis, and bioimpedance. The duration of intervention was based on minimizing logistic issues that could risk the quality of data collection and creating a higher rate of adherence to the project. Children and teens wore a “body bug” during several school days that measured motion, steps, skin temperature, galvanic skin response, and calculated basal metabolic rate. Table 1 lists the variables measured for this study. Food access and affordability were assessed at 41 food markets in the same geographic area as the three schools. Data were also obtained from public health databases for the state, region, and city. The Harmonized Micronutrient Project is registered at ClinTrials.gov (#NCT01823744) and was carried out according to the Helsinki Declaration.

### 2.2. General Statistical Analysis

The reluctance to embrace n-of-1 study designs is usually expressed as (most simply) “how do you analyze individual level data?” Recent publications have provided guidance on various n-of-1 designs (e.g., [14,15,16,17,18]) that account for randomization/counterbalancing, blinding, replication, carryover effects, adaptation, multiple outcomes, and multiple subject designs [15], including power calculations for determining sample size [16].

Aggregating data from all participants at each time point also provided the status of vitamin levels in the population. All statistical analyses were performed with R statistical software version 3.0.1 and with Statistical Package for the Social Sciences (SPSS), version 20.0. Nonparametric Mann–Whitney and Kruskal–Wallis tests were used for between-group and -cohort comparisons, respectively. Spearmen’s rank correlation test and Wilcoxon’s signed rank test were also considered when needed. Chi-square tests were used to compare distribution of categorical variables. Analysis of variance (ANOVA) for repeated measures was used for longitudinal analyses and linear regression covariance analyses (ANCOVA) to compare groups, with adjustment for confounding variables. Nominal associations were set to 5% for unadjusted *p*-values. When specified, *p*-values were corrected for multiple testing, calculating the false discovery rate (FDR) of adjusted *p*-values using the Benjamini-Hochberg procedure, and the significance threshold was set to 0.1.

To assess the significance of the micronutrient intervention at the population level, we also used a method that accounted for effect size above regression to the mean (RTM) [19], which may occur in situations of repeated measurements when extreme values tend to regress to the population mean. Almost all measured metabolites displayed inter-individual variation before and after the intervention, but there were insufficient numbers in subgroups (aggregating those who responded similarly) to understand those responses. Hence, in addition to the n-of-1 analysis, population-level data were also analyzed through aggregating data at each time point and in both years. The replication arm tested the results obtained in the original study. The prediction model of response to intervention was performed using first-year data as the training set, the second year as the test set, and elastic net regression in the *glmnet* 2.0-7 R package [20].

## 3. Primary Results [21]

### 3.1. Population-Level Results

While the primary end points of this study were to analyze the response of individuals who consumed supplemental multiple micronutrients for 6 weeks, results of analyses of baseline anthropometric, clinical, and omic data defined the nutritional status of the population. With a few exceptions, such as age and tanner score, and some variables that could not be readily explained (calcium, iron, phosphate, mean corpuscular volume, mean corpuscular hemoglobin, basophils), the two populations were metabolically similar. The primary results included the following:The studied population in two years (*n* = 280) had an average age of 11.7 ± 1.1 years old, 55% were female, 43.2% were overweight or obese, and 73.6% were at pubertal status 2 and 3.The average total Brazilian healthy-eating index (BHEI-R) score for all included individuals was 54.8 ± 7.5 (53.7 in the first year and 54.5 in the second year). One hundred and fifty-two participants in 2013 and 2014 (91%) had a total BHEI-R below 65, which is considered a “poor diet”, specifically poor in vegetables, legumes, fruits, whole grains, milk and dairy, and rich in sugar and saturated fat; 15 (9%) were classified in the intermediary category, with scores between 65 and 84; and none were in the “good diet” category (above 85) [22]. No statistical differences were found in food-intake patterns across all three visits.The percent *insufficiencies* (in parenthesis) obtained by aggregating data from all participants in both arms of the study were in folate (43%), retinol (24%), α-tocopherol (25.8%), γ-tocopherol (100%), thiamine (99%), vitamin B12 (43%), nicotinamide (8%), pantothenic acid (99%), and pyridoxal (76%).A total of 16% of the population were classified as having dyslipidemia.The blood level of thymidine monophosphate was positively correlated with percentage of European ancestry, while the levels of vitamin B12 and folate were negatively correlated with percentage of Native American ancestry.

#### Relevance

The United Nations Children’s Fund (UNICEF), United Nationals University (UNU), and the World Health Organization approved the United Nations International Multiple Micronutrient Antenatal Preparation (UNIMMAP) for pregnant women in low- and middle-income countries. UNIMMAP contained vitamin A, vitamin B1, vitamin B2, niacin, vitamin B6, vitamin B12, folic acid, vitamin C, vitamin D, vitamin E, copper, selenium, iodine, iron, and zinc, and it reduced the incidence of low birth rate (rev in [23]). While the socioeconomic and environmental setting and participant life stage and age differed, the 9–13-year-old participants in this study had a high prevalence of being overweight and obese and dyslipidemia, which is consistent with the measured poor dietary intakes. Others have also found similar results in other global populations in this age group. The aggregated baseline data demonstrated that high percentages of the participants in our study had micronutrient insufficiencies or deficiencies, but individuals varied in levels of plasma micronutrients. These data support our approach to use a multi-micronutrient intervention that was not based specifically on the levels of micronutrients of the population in our study.

### 3.2. Population-Level Analysis of Clinical and Omic Measures Post-Intervention

The availability of data from the replication arm tested the first-year results. The levels of many blood or red-blood-cell metabolites were reproducibly changed at stringent statistical levels (at least *p* < 0.05). The intervention decreased levels of albumin, basophils, calcium, glucose, low density lipoprotein (LDL)-cholesterol (Figure 1), γ-tocopherol and increased levels of mean corpuscular volume, flavin mononucleotide (FMN), nudifloramide, pantothenic acid, pyridoxal, α-tocopherol, 5-methyltetrahydrofolic acid, folate and vitamin B12 in both years, according to the regression of the mean test [19], indicating that these metabolites were consistently changed by the intervention. Magnesium, iron, phosphorus, riboflavin, 25-hydroxy-vitamin D3, flavin adenine dinucleotide (FAD), nicotinamide (NM), pyridoxic acid, retinal, and α/β-carotene were not statistically different after the intervention.

HDL cholesterol, triglycerides, and VLDL-cholesterol also showed no differences. These results indicated that the vitamins and minerals in the supplement produced targeted changes in physiological processes rather than an overall system-wide effect.

#### Relevance

The combination of multi-micronutrients used for this intervention reproducibly increased or decreased the levels of circulating forms of nine organic vitamin metabolites and three clinical variables, as expected. Decreases in average levels of blood lipids and glucose suggest that one or a combination of vitamins and minerals in the supplement influenced metabolism. A multi-micronutrient (with 24 vitamins, minerals and metabolites) intervention in adults (age 24–79) lasting 24 weeks did not find changes in the lipid profile of glucose levels but did alter LDL oxidation indices [24]. However, changes in levels of various plasma lipids in response to single- or multiple-micronutrient supplementation were observed [25,26,27,28,29,30,31,32,33,34,35]. The results of our study suggested that changes in the circulating levels or bioavailability of a combination of micronutrients interacted with cholesterol pathways and networks, and led to the observed changes in plasma levels of total cholesterol and LDL-c. Although these results require and deserve further investigations, the results suggest that multiple-micronutrient supplementation may positively affect metabolic syndrome in children and adolescents.

### 3.3. Inter-Individual Variability

Data aggregated from all participants at each time point demonstrated that the intervention altered the levels of numerous metabolites at the population level (e.g., LDL-C Figure 1). However, metabolite responses were individually unique among study participants. For example, we used the clinical daily variation (+/−~10%) of LDL-c [36] to identify three response groups (Figure 2). Many individuals decreased LDL-c, a second smaller group increased LDL-c, and the majority either decreased or increased LDL-c less than 10%. This overall pattern was repeated in the replication arm. Only about 50% of individuals who participated in the replication experiment responded in the same direction as the first year for virtually any measured variable. We attributed this finding to age stage: 9- to 13-year-old individuals experience growth and sexual maturation which may alter responses to nutrition over the course of a year. Although this study was well powered for many of the variables tested, there were insufficient numbers to conduct subgroup analysis to explain why these groups differed.

In addition, identifying the three LDL response groups (metabotypes public domain pathway databases DAVID and Metacore) could link control of LDL catabolism through the liver X receptor with six replicated plasma vitamins analyzed in this study.

#### Relevance

Each individual responded to the intervention uniquely, a result consistent with the long-recognized biochemical individuality and responses to nutrition summarized by Williams in 1956 [37]. Inter-individual responses complicate nutritional interventions designed for the “average” person in a population. Two recent publications reviewed studies examining the effects of single or multiple SNPs on plasma triglyceride in response to omega-3 intake [38] and cholesterol levels in response to diets [39]. We did not evaluate genetic variation that may have contributed to the inter-individual response in response to the micronutrient intervention. A challenge for basic and public health researchers is to identify easily-tested biological markers associated with responses to micronutrients that account for inter-individual variation in diet intakes, genetic makeup, environment, and metabolic physiology [40].

### 3.4. Predicting Responses to the Intervention

A primary goal for personalizing nutrition is to predict individual responses to a nutritional change. Using elastic net regression (Enet [20]), we explored whether multiple variables and baseline values could explain the variation in clinical and metabolite endpoints. The variable most predictive of response for most metabolites was the baseline value: low values predicted a strong response and higher baseline values predicted less of a response. Enet results also identified other metabolites that contributed to the levels of nicotinamide, γ -tocopherol, and iron. An example was nicotinamide, whose levels were positively affected after intervention by higher baseline values of platelets, calcium, iron, and retinol. In addition, lower baseline numbers of vitamin D, age, 5-methyltetrahydrofolate, and α-tocopherol produced higher nicotinamide levels after intervention.

#### Relevance

The experimental design provides a potential strategy for identifying metabolites or blood components that contribute to responses to nutritional changes and, therefore, guidance to improve health status. Our report contributes to an emerging trend of precision nutrition, using physiological measures to predict response to diet. Others have used postprandial glucose levels [6], k-means groups based on plasma triacylglycerol, total cholesterol, HDL-cholesterol, and glucose [41], and response to meal challenges (rev in [13]) to predict responses to dietary changes. The microbiome is also likely to contribute to individual responses to nutrition (rev. in [42,43]). Although precision nutrition has generated over a thousand of publications (1305 citations in PubMed as of 28 December 2023), no single strategy has emerged to predict an individual’s response to a diet.

## 4. Secondary Results

### 4.1. Food-Intake Studies

Accurate assessment of dietary exposure is crucial in investigating associations between diet and disease. Quantifying food consumption using surveys has the disadvantage of being unable to measure changes in the concentrations of nutrients and bioactive compounds in food. Furthermore, measuring food consumption does not necessarily account for the bioavailability of nutrients, inter-individual variability in their absorption, digestion, and processing—processes that result from the interaction of an individual’s genetic makeup and interaction with local environmental factors [44]. Biological markers have a promising role in validation studies of traditional food-intake methods since they could be an objective measure of exposure to food. However, biological markers identified through samples in cohort studies usually do not demonstrate a dose response to consumption. In addition, in these calculated exposures blood-biomarker associations should be considered as putative biological markers that require further validation [45]. The two following sub-studies are aimed at finding putative biological markers for nutrient and food groups.

#### 4.1.1. Healthy-Eating-Index Biomarkers [46]

Our diet-related-biomarker sub study [46] first assessed the Revised Brazilian Healthy Eating Index (BHEI-R) as a measure of dietary status in 167 Brazilians children and adolescents, aged 9 to 13 years, and secondly determined the strength of the correlations between BHEI-R scores and plasma concentrations of a subset of plasma vitamins, fatty acids, and one carbon cofactor with the results shown in Table 2.

#### 4.1.2. FFQ Biomarkers

A second sub-study [47] evaluated the validity of nutrient and food-group intake estimated by an FFQ against biomarkers in 210 Brazilian children and adolescents aged 9–13 years. Intakes were correlated with biomarkers in plasma and red blood cells (Table 3). In general, BHEI-R and the FFQ used in our study are valid tools for ranking dietary intake of several nutrients and food groups in children and adolescents aged 9–13 years living in southeast Brazil. Although biomarkers were associated with nutrients and food groups from different food-intake methods, some results are consistent between the two sub-studies, such as (i) vegetable intake and β-carotene, (ii) animal protein intake and creatine, and (iii) milk and dairy intake with pyridoxal 5′-phosphate, which indicates an important step in the validation of these metabolites as food-intake biomarkers in future studies.

#### 4.1.3. Relevance

This study was the first to demonstrate the use of a BHEI-R assessment tool in a pediatric population and its validation with biomarkers associated with diet quality and healthy-food-intake patterns. FFQ proved to be a valid tool for ranking the dietary intake of several nutrients and food groups in children and adolescents aged 9–13 years living in southeast Brazil and will be useful in further studies analyzing diet and health outcomes in Brazil. Biomarkers have been found for plants in adults [48] and children/adolescents [49], and for meat and seafood [50], fermented products [51], spices [52], cereals [53], and allium vegetables [54]. Collectively these biomarkers will enable the development of diagnostics for determining adherence to personalized diets.

### 4.2. Impact on Lipid Species [55]

The changes in the LDL-c [21] led to a more complete analysis of lipids in blood samples from participants in the 2013 year only (due to a batch effect in the proteomic analysis in year 2). Specifically, 1129 plasma proteins using the SOMA scan platform (Soma-Logic, Boulder, CO, USA), as described [56]. Lipid extraction and analysis of other metabolites were performed as reported previously [55]. We conducted untargeted lipidomics of intact lipid species and identified DAGs (diacylglycerols), SEs (sterol esters), TAGs (triacylglycerols), free cholesterol, LPCs (lysophosphatidylcholines), LPAs (lysophosphatidic acids), LPEs (lysophosphatidylethanolamines), PIs (phosphatidylinositols), and PCs (phosphatidylcholines), among others. We analyzed the impact of micronutrients on individual lipid species of the plasma lipidome and proteome. Proteomic analysis was conducted using DNA aptamer technology.

Overall, we observed statistically significant reduction in lipid levels after the intervention and following the washout (Appendix A). Fifteen lipids were reduced significantly between baseline (Visit 1) and post-intervention (V2), suggesting a direct and rapid impact of intervention. These included 6 SEs, 3 PIs, 2 PCs, 2 LPCs, free cholesterol, and 1 LPE. SEs’ levels were reduced the most (7.49–12.67%) based on mean fold changes. Overall, correlation analysis highlights the potential role of fat-soluble vitamins (e.g., A, E, α, and β-carotene), and their potential interactions with different lipids classes, specifically SEs, PCs, TAGs, DAGs, and PIs. But after intervention (V1–V2), only α-tocopherol, an antioxidant vitamin, was highly correlated to several PCs, PIs, TAGs, SEs, and DAGs (Appendix A). These results are shown schematically in Figure 3. α-tocopherol levels improved after the intervention, and it was one of the vitamins that passed the RTM test described in our previous publication [21].

We identified approximately 30 proteins related to lipid metabolism which were specifically impacted by the intervention (based on statistically different levels between V1 and V2; *p* values after FDR correction < 0.05). These proteins are involved in phospholipid metabolism; glycerophospholipid metabolism; sphingolipid de novo biosynthesis and metabolism; synthesis of PA/PC; and acyl chain remodeling of PI/PG/PS/PC/PE. Phospholipases (PLA2G2A, PLA2G10, PLA2G5), which mediate hydrolysis of phospholipids to fatty acids, were changed using the intervention. For example, PLA2G2A (phospholipase A2 group IIA) levels increased following the supplementation (*p* = 0.0108). In contrast, PLA2G10 (phospholipase A2 group 10, *p* = 0.00004) and PLA2G5 (phospholipase A2 group V, *p* = 0.0058) levels decreased after the supplementation (*p* = 0.00004).

The fifteen lipids reduced between V1 and V2, suggested a direct and rapid impact of intervention. The 30 proteins found using proteomic analyses were involved in phospholipid metabolism, glycerophospholipid metabolism, sphingolipid de novo biosynthesis and metabolism, synthesis of PA/PC, and acyl chain remodeling of PI/PG/PS/PC/PE. Specific details are described in [55].

#### Relevance

Vitamins A [57], B12 [58], B6 [59], and D [26] are involved in fat metabolism in vitro and in vivo, although these studies did not analyze the sterol esters and phospholipids measured in this study. A review described the effects of macronutrients and niacin, copper, zinc, magnesium, and calcium on post-prandial lipemia [60]. The results of our sub-study suggest that the multi-micronutrient supplementation had combinations of short-term and long-term impacts on lipid metabolism with a significant effect on the circulating levels of phospholipids, lysophospholipids, and cholesterol esters. A correlation-based analysis highlighted the potential importance of α -tocopherol (vitamin E), in comparison to the other components in the supplement, in changing lipid species between baseline and post-intervention. The changes in levels of these lipid species suggest that micronutrients may play a role in reducing the risk of cardiovascular diseases.

### 4.3. Lipoproteins, Polyunsaturated Fatty Acids (PUFAs), 1-Carbon Pathway Metabolites, and B Vitamin Associations

We developed a high-throughput LC/MS method to analyze 13 cofactors and metabolites in the one carbon pathway [61] which included vitamins B2, B12, B6, folate, homocysteine (Hcy) S-adenosylmethionine (SAM), and S-adenosylhomocysteine (SAH). SAM is the methyl donor for ~200 methyltransferase reactions involving DNA, RNA, protein, and metabolites [62,63]. Plasma levels of SAM, SAH, and homocysteine have independently and in combinations been associated with immunological, obesity, cardiovascular, and neurological pathologies ([64,65] and references therein). Previous research from our team analyzed the methylation potential in children attending a summer day camp [64] and subsequently associated the SAM:SAH ratio with immunological and gastrointestinal networks [65].

#### 4.3.1. Vitamins and One-Carbon Metabolites [66]

A subset of the associations between B vitamins, PUFAs, and 1C metabolites (Table 4, from [66]) and B vitamins and PUFAs were found in this micronutrient-intervention study (Table 5). Results of the intervention linked B2 and folate with improvements in fatty acids (FA—Table 5). An increase of B6 and B12 values decreased homocysteine levels (Table 4). These findings indicate the need for further studies of the role of B vitamins in the metabolism of lipoproteins and FA in children.

#### 4.3.2. Lipoproteins, PUFAs, and B Vitamin Associations [66]

This micronutrient-intervention study generated diverse data sets that allowed for discovery-based analyses of the effects of micronutrients on different physiological outcomes. The changes in LDL-cholesterol (Figure 2) and minor lipid components of plasma and red-blood-cell membranes [21,55] led us to analyze potential associations between plasma levels of B vitamins, lipoproteins, and polyunsaturated fatty acids (PUFA) species and, subsequently, the correlations between PUFAs, vitamins, and DNA damage.

None of the B-vitamins alone were associated with high levels of HDL-c and low levels of total cholesterol, triglycerides, or LDL-c suggesting that other components of the intervention produced “healthy” lipid profiles, such as α-tocopherol, as mentioned before. Somewhat unexpectedly, vitamin B2 and B6 were positively associated with total cholesterol and LDL-c: an increase of 1 nmol/L in vitamin B2 was associated with an increase of 0.09 mg/dL of LDL-c and 0.05 mg/dL of total cholesterol. These results may explain the responses of some individuals with increased LDL-c following the intervention (Figure 2, orange bars).

B vitamins were associated with changes in levels of many red-blood-cell-membrane fatty acids and soluble metabolites (Table 5) as well as RBC MUFAs and SFAs levels [66]. B vitamins had associations specific to fatty acids (Table 6).

#### 4.3.3. Relevance

Although evidence of associations between B-vitamins and FAs is unclear and often debatable (e.g., [67]), these findings indicate that an adequate intake of vitamin B2 and folate may improve the FA profile. Appropriate concentrations of *n*-3 and *n*-6 FA contribute to the development of the central nervous system and renal health and to the prevention of CVD [67], which are increasingly common due to a sedentary lifestyle, poor diet, and excess body fat in children and adolescents. The prevention of CVD risk factors should begin even in childhood since habits for life are formed at this stage.

### 4.4. PUFAs and DNA Damage [68,69]

We found an inverse relationship between DNA damage as measured by the comet assay (alkaline single-cell gel electrophoresis) [70,71] and blood levels of specific PUFAs [68]. Two clusters (*n* = 62 and *n* = 78) were identified based on the extent of tail intensity (5.9% +/− 1.2 versus 13.8% +/− 3.1, respectively). EPA and DHA, but not retinol, riboflavin, or beta-carotene, were higher in the low-DNA-damage group compared to the high-DNA-damage group. EPA and DHA are substrates for pro-resolving mediators of inflammation [72].

We subsequently analyzed the association of 117 plasma proteins involved in inflammatory processes and DNA damage [69]. These proteins were selected from the 1129 plasma proteins quantified with Somascan (version 1—Somalogic, Inc., Boulder, CO, USA) and analyzed against results of the comet assay. Six of the 117 proteins were statistically different between the two groups. Cyclin-dependent kinase 8 (CDK8), cyclin C (CCNC), kynureninase (KYNU), phosphatidylinositol 3-kinase catalytic subunit alpha (PIK3CA), phosphatidylinositol 3-kinase regulatory subunit 1 (PIK3R1), and protein kinase C beta (PRKCB) were higher in the high-DNA-damage group which also had lower levels of DHA and EPA compared to the low-DNA-damage group. These results have potential long-term implications since low DHA and EPA and elevated inflammatory proteins contribute over time to increased DNA damage and its consequences for disease development. We could not confirm these findings in the replication arm due to an unexplained batch effect at two of the three timepoints.

#### Relevance

The association of specific micronutrients (retinol, riboflavin, β-carotene) and DNA damage/repair has been intensively studied [73,74]. Increased DNA damage has been associated with cancer [75], immune dysfunction [76], and age-related diseases (e.g., [77]). Our findings show that children and adolescents with higher DNA damage and lower levels of n-3 PUFAs, that is, DHA and EPA, presented higher levels of proteins involved in inflammatory mechanisms. The importance of the results obtained in this sub-study include the potential for long-term negative impacts of these pro-inflammatory proteins on DNA-damage recurrence in children and adolescents with low DHA and EPA levels. DNA damage is a crucial component in the development of several diseases.

### 4.5. Metabo Groups

The classification of individuals into subgroups according to their metabolic profile is defined as metabotyping, and this approach has been employed to successfully identify differential response to dietary interventions in population groups [78]. The application of metabotyping in longitudinal studies demonstrates that metabolic groups (metabotypes) can be associated with cardiometabolic risk factors and diet-related diseases [79], while its application in interventions can identify metabotypes with different responses. Nutritional advice based on metabotyping may be more effective in promoting improvements in dietary habits when compared to advice based on population averages [80]. Two metabotypes were identified in our study based on (a) lipid profiles to characterize different responses after daily supplementation of vitamins and minerals [81] and (b) based on vitamin levels to better characterize metabolic groups in individuals with a greater need for nutritional counseling [82].

#### 4.5.1. Lipid Profile and Proteomic Metabotypes

Twenty randomly selected participants were divided into two metabolic groups according to their lipid profiles ([81] and Table 7). Even though there were no differences in demographic variables between groups, micronutrient supplementation differentially improved plasma vitamin levels: pantothenic acid (vitamin B5), pyridoxal (a form of vitamin B6), and pyridoxic acid (a catabolic product of vitamin B6) plasma levels increased in pool 1 (better lipid profile compared to pool 2), but α-tocopherol and pantothenic acid improved in pool 2. The lipid profile improved only in individuals in pool 2 with a decrease in LDL from V1 to V2 in response to the micronutrient supplementation. In this sub-study analysis, increases of pantothenic acid, pyridoxal, and α-tocopherol after supplementation were consistent with previous results [21,55]. This subset analysis further supported the finding that individuals respond differently to an intervention.

Multiple regression analysis found that an increase in the percentage of Native America genetic ancestry and differences in sex can, in combination, predict 29% of the fold-change variation for α-tocopherol from visit 1 to visit 2 (r = 0.62; R^2^ = 0.38; adjusted R^2^ = 0.29; ANOVA *p* = 0.007).

iTRAQ proteomic analysis requires larger sample sizes, so six sample pools were analyzed, a sample pool for each group at each time point. Twenty plasma proteins were identified using proteomic analyses that changed expression after micronutrient supplementation in at least one of the pools, and 18 presented a fold change ratio of ≥1.20 or ≤0.80. Most of the identified proteins had different levels among the pools after the intervention. Briefly, the expression of alpha-1 antitrypsin, haptoglobin, Ig alpha-1 chain C region, and plasma protease C1 inhibitor increased in pool 2. These proteins are associated with positive physiological effects in the lipid/glucose metabolism, micronutrient transport/metabolism, and in the immune system [83,84,85]. Expression of alpha-1-acid glycoprotein 1 and fibrinogen alpha-, beta-, and gamma-chains decreased in response to the intervention in pool 2 which are beneficial for health. A decrease in levels of these markers after intervention may benefit individuals, since they have been associated with an improved acute phase [86,87,88] and/or cardiovascular health [89,90,91]. Individuals in pool 1 did not show any improvements in lipid profile, and the analyzed proteins responded inversely or did not change their levels.

#### 4.5.2. Vitamins—Inflammatory Biomarker Metabotypes

Over 40% of this study population were overweight or obese [21]. Unbalanced nutrition can contribute to a state of low-grade inflammation during childhood, especially in those who are overweight (e.g., [92]). Certain B vitamins and vitamin A may be involved in inflammatory pathways associated with homocysteine (e.g., [93]) and leukotriene A4 hydrolase (LTA4H) [94], which have been considered independent predictors of low-grade inflammation, atherosclerosis process, and all-cause mortality.

κ-mean cluster analysis found the dietary components of vitamins, fatty acids, and plasma metabolites to be related to vitamin levels and inflammatory markers (Table 8 from [82]). The metabolic group with low plasma levels of B-vitamin biomarkers (riboflavin, pyridoxal, and vitamin B12) also had a lower dietary intake of B-vitamins and higher RBC homocysteine levels. In clinical practice, these individuals should receive nutritional counseling. The levels of riboflavin, pyridoxal, cobalamin, and homocysteine levels predicted 9.0% of LTA4H variation in the total studied population, although LTA4H was not statistically different between metabolic groups. Further studies are needed to elucidate the role of these B vitamins in regulating LTA4H, but this pilot study indicated that these metabotypes may be of use for nutritional counseling.

#### 4.5.3. Relevance

Virtually all plasma metabolites measured in the studies reviewed here were similar to the pattern observed for LDL-cholesterol (Figure 2); that is, there was a “continuous” distribution of the measured metabolite across the population. As a step toward personalized nutrition, we and others are defining “metabolic groups”—or metabotypes [95] (and rev in [79])—as based on some a priori criteria [41] or through discovery-based approaches using unsupervised algorithms. Metabotypes group individuals with similar metabolic, genomic, or proteomic similarities.

The first metabotypes groups revealed proteins that are associated with positive physiological effects in the lipid/glucose metabolism, micronutrient transport/metabolism, and in the immune system [83,84,85] and were differentially expressed in metabolic different individuals. These individuals will need different nutritional counseling approach proving that one size does not fit all.

The metabolic group identified in the other substudy, which had low plasma levels of B-vitamin biomarkers (riboflavin, pyridoxal, and vitamin B12), lower dietary intake of B-vitamins and higher RBC homocysteine levels, needs nutritional counseling. To prevent or delay onset of noncommunicable diseases, B-vitamins, homocysteine, and LTA4H should be monitored in children and adolescents. Further studies are needed to elucidate the role of B-vitamins in regulating LTA4H.

### 4.6. Identification of Vitamin B12 Genetic-Risk Score [96]

One of the goals of precision nutrition and health is to understand the genetic contribution made to the physiological traits of individuals. Our initial genetic analysis [21] revealed that plasma vitamin B12 levels were negatively associated with increasing Native American ancestry. We subsequently used a middle out approach to identify single-nucleotide polymorphisms in 90 genes associated with plasma levels of vitamin B12. Of a total of 3,406,465 SNP variants analyzed using whole-genome mapping and whole-exome sequencing, 6999 were present in the 90 B12-related genes. Following LD clumping, a generalized estimated equation identified 36 SNPs weighted by effect size in 26 genes associated with vitamin B12 levels at baseline. Since population architecture can affect SNP-condition associations [97], the efficient local ancestry inference (ELAI—[98]) method was used to account for African, Native American, and European ancestry at each SNP in each individual. EUR-derived SNPs were positively correlated with baseline vitamin B12 levels, while AFR and AMR ancestries were negatively correlated. These associations corroborated the observed negative association with AMR in Mathias et al.’s study. These 36 ancestry-corrected SNPs from an additive model constitute a polygenic risk score (PRS) for vitamin B12, which explained 42% of the baseline plasma vitamin B12 levels [96].

Even though the genes preselected to create the PRS were involved in vitamin B12 metabolism and function, we used String [99] to visualize potential functional interactions among the 26 genes contributing to the GRS:LMBRD1 (lysosomal cobalamin transport escort protein), CUBN (cubilin), TCN1 (transcobalamin 1), TCN2 (transcobalamin 2), and ABCD4 (lysosomal cobalamin transporter ABCD4) are involved in the transport of or in the lysosomal release of vitamin B12 into the cytoplasm.CUBN, ABCD4 with MTR (exosome RNA helicase) participate in reactions that catalyze the transfer of a methyl group from methyl-cobalamin to homocysteine.A small interconnected lipid metabolism network included (i) NDUFAB1 (mitochondrial acyl carrier), an acyl carrier protein of the growing fatty acid chain in fatty acid biosynthesis, (ii) MVK (mevalonate kinase), a regulatory site in the cholesterol biosynthetic pathway, (iii) PEMT (phosphatidylethanolamine N-methyltransferase) which catalyzes the three sequential steps of the methylation pathway involving phosphatidylethanolamine (PE), phosphatidylmonomethylethanolaimne (PMME), phosphatidyldimethylethanolamine (PDME), phosphatidylcholine (PC), and SLC27A4, a fatty-acid transport protein. These interactions are consistent with the changes in lipidemia ([55] and Lipidemia section).Vitamin B12 levels may also have a role in the vitamin D metabolic processes through the low-density lipoprotein receptor-related protein 2 (CUBN, LRP2) and in a-amino acid metabolic process (CBS, FPGS, MTR, PEMT, SARS).

A key goal of this micronutrient project was to translate research findings to practical applications. Hypothetically, nutritional counseling may be based on PRS terciles (Figure 4), nutritional, and demographic data to prevent insufficiencies/deficiencies or to select interventions as suggested for other PRS applications to clinical settings [100]. In addition, nutritional counseling based on PRS and considering cofounding variables may individualize management of vitamin B12 recommendations and personalize health care (Figure 5).

#### Relevance

Thousands of studies have attempted to associate a single-nucleotide polymorphism (SN) with levels of plasma metabolites and/or response to diet [101], but in all but a few exceptions (e.g., lactase persistence [102]), the effect size is negligibly small despite being statistically significant (rev. in [103]). However, combinations of SNPS could explain a significant portion of the variation in phenotype incidence in the general population. A polygenic risk score (PRS or alternatively PGS)) aggregates the effects of many genetic variants into a single number which predicts genetic predisposition for a phenotype. The use of PRS is becoming clinically highly useful—over 622 traits now have PRS in the PGS database (https://www.pgscatalog.org/—accessed on 29 December 2023).

The SNPs identified in this study may contribute to variations in B12 levels. We propose a hypothetical nutritional counseling based on PRS terciles, nutritional, and demographic data, and assuming that PRS can play a role in helping to screen for and prevent insufficiencies/deficiencies or to select interventions. In addition, nutritional counseling based on PRS and considering cofounding variables may individualize management of vitamin B12 recommendation and personalize health care.

## 5. Main Results, Strengths, and Limitations

Summarized here are key findings and limitations of The Harmonized Micronutrient Intervention study. We refer the reader to full articles and specific sections for discussion of the relevant literature. This study was designed as n-of-1, discovery-based and translational research project that analyzed the effects of 12 vitamins and five minerals at near to 100% of daily intake levels on multiple physiological systems [21]. Multiple micronutrients are usually provided to pregnant women in low-income settings to reduce the incidence of low birth rate (rev in [23]). We reasoned that the micronutrient status of individuals in any population may not reflect the population averages reported in the literature or public-domain databases (e.g., National Health and Nutrition Examination Survey—NHANES).

The intervention decreased levels of albumin, basophils, calcium, glucose, LDL-cholesterol, γ-tocopherol, and increased levels of mean corpuscular volume, FMN, nudifloramide, pantothenic acid, pyridoxal, α-tocopherol, 5-methyltetrahydrofolic acid, folate, and vitamin B12 at the population level in both years [21]. The changes in the LDL-c, fatty acids, and in 15 other lipids suggest a direct and rapid impact of the micronutrient intervention [55]. The improvement of α-tocopherol was highly correlated to several lipid profile improvements in a subset of the population. Pantothenic acid, B6, B2, and folate were also linked to the improvement of fatty acids in another sub-group, emphasizing the importance of considering metabotypes for nutritional counseling [66]. The supplement by itself improved plasma levels of these vitamins, possibly corroborating the nutrient–nutrient interactions in improving the metabolic profile. We also identified 36 SNPs weighted by effect size in 26 genes associated with vitamin B12 levels at baseline [96].

A primary feature of the experimental design was the n-of-1 approach (discussed in [104]) with each participant having his/her own control accounting for inter-individual variability and a replication arm that repeated the same intervention one year apart. A strength of the study was that graduate students associated with the project assured that the students swallowed the tablets with 98% compliance. Assessing the same biochemical, clinical, and physiological measures provided insights as to which omic technologies could produce reproducible results at the group level, with the caveat being that the physiology of participants may change due to age-related developmental transformations. Most clinical and omic data were reproducible at the population level (i.e., aggregating data at each time point) with the exception of the DNA-damage assay and the Somalogic proteomic data. In both cases, we observed batch effects at different time points in different years restricting the analysis to only results from the first year of the study. Complete blood counts and clinical biomarkers are routinely conducted over time for adults, but not typically in the age range of this study (9 to 13 years old). We are unaware of other nutrition research studies with a replication arm.

The study was designed to further the development of strategies and methods to enable precision nutrition and health. A major goal of that initiative is to predict response to dietary changes (rev in [105]). The results of this study show that the best predictor of individual response to any variable was its baseline level [21]. That is, the change in levels of any given metabolite was greater if the metabolite had a low level at baseline, and lower if the metabolite was at a high plasma level. Nonetheless, other measured variables contributed to the change in specific metabolite levels; the best example was the contributions of baseline values of platelets, calcium, iron, and retinol to the level of nicotinamide. Such findings can be leveraged to better understand the processes that produce individual responses to nutritional changes. However, studies with larger numbers of participants are necessary to confirm and extend these results.

This project followed tenets of community-based participatory research methods (translational research, e.g., [11,106,107]). Members of our team met with principals and staff at eight schools in Ribeirao Preto, with three choosing to participate. Subsequent community meetings were held at each school with parents and students to explain the research program and to solicit input about health concerns of the community members. Parents of each participant were present at assessments and contributed with information on dietary questionnaires and other behavioral activities.

The research staff also conducted four health fairs with the participants and their families after the 2-year project was completed. Each health fair presented overall population-level reports on specific topics (e.g., clinical biochemistries or genetic ancestry) and each participant with a parent received their individual results in the presence of a certified clinical nutritionist.

The limitations of the studies were (i) the small sample size with a context specific to urban, middle-class culture in Brazil and a unique genetic admixture, which makes it difficult to extrapolate the results to the entire population of children and adolescents; (ii) an absence of important indicators of metabolites’ status, such as levels of methylmalonic acid and holotranscobalamin for vitamin B12 (the fraction available for tissue uptake); (iii) the use of pooling samples in one of the proteomic studies (Lipid Profile Metabo Group study [81]), which eliminated the possibility of analyzing samples individually or testing the association of vitamin levels, proteomic data, and ancestry; and (iv) the batch effect in the comet assay and the proteomic analyses in year 2, which precluded replication for those data sets.

## 6. Conclusions

The intent of this research project, initially designed in 2011, was to conduct a nutritional-challenge study driven by physiological, biochemical, behavioral, dietary, environmental, and demographic data (as per [108,109]) with discovery-based statistical and algorithmic methods to identify key variables to explain responses to a multiple-micronutrient intervention. Given the constraints of training PHD candidates and postdoctoral fellows (who each require their own, targeted research project) and our current publication practices (articles are usually focused on single topics), this research project contributed to understanding how physiology (specially lipid profile) changed in response to vitamins and minerals supplemented in a sub-group of children and adolescents that had a very poor dietary pattern according to the Brazilian Healthy Eating Index, specifically, poor in vegetables, legumes, fruits, whole grains, milk and dairy, and rich in sugar and saturated fat.

Since current political, public health policy, and behavioral habits are inadequate for providing and supporting accessibility to healthy foods, the use of multi-micronutrient supplements that are below upper tolerable levels may complement Brazilian 9- to 13-year-old individuals’ needs and improve their metabolic profiles. This research project showed the importance of gathering diverse data sets (e.g., including both omics and features derived from unstructured data) avoiding unmeasured confounding data and adding the possibility of adding covariates in statistical analyses. Additional studies with similar designs are needed in different populations with similar data sets to test the results obtained from the project and to confirm interpretations. Overall, we propose that the judicious application of big data in nutrition science could offer more tools to understand the complexity of nutrition and its effect on maintaining health and delaying or preventing the onset of chronic diseases.

## Figures and Tables

**Figure 1 nutrients-16-00188-f001:**
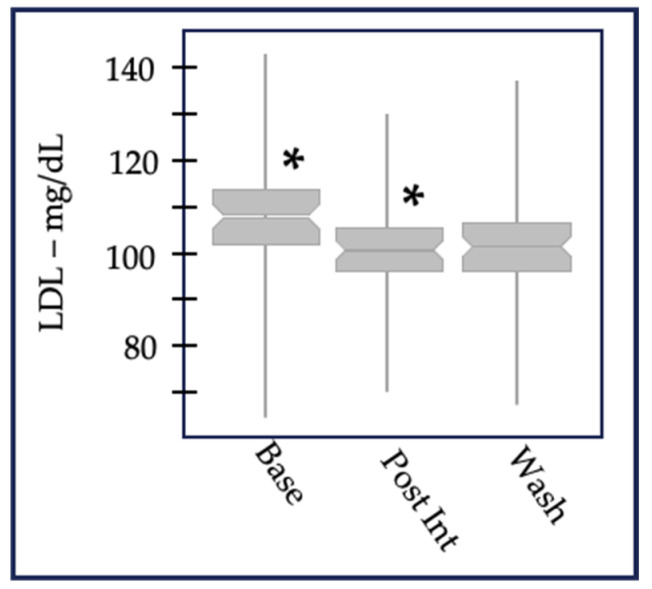
LDL-cholesterol values aggregated at baseline, after 6 weeks of post-intervention, and after a 6-week washout. The change between baseline and post-intervention was statistically significant (designated by *) in both 2013 (*p* = 1.4 × 10^−6^ for *n* = 98) and 2014 (*p* = 3.4 × 10^−3^ for *n* = 107). LDL-c, like other lipid metabolites measured in this study, did not return to baseline values after 6 weeks, suggesting there were long-term effects of the multiple-micronutrient intervention.

**Figure 2 nutrients-16-00188-f002:**
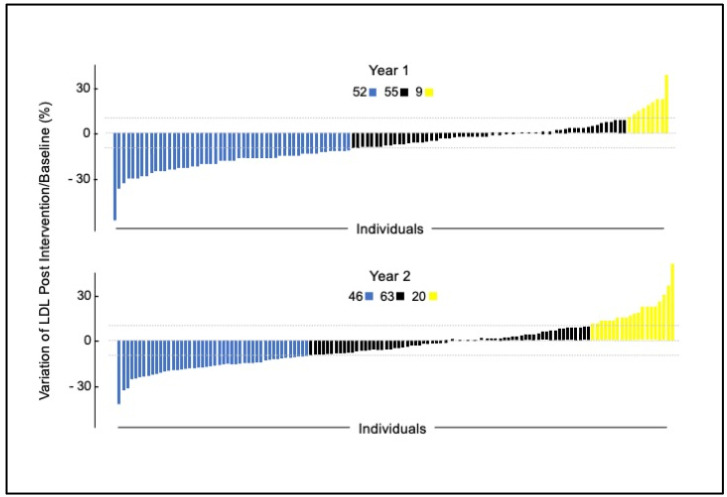
Individual clinical daily variation of LDL-c from timepoint 1 to timepoint 2 in 2013 and 2014 after intervention. Dotted lines indicate (+/− ~10%).

**Figure 3 nutrients-16-00188-f003:**
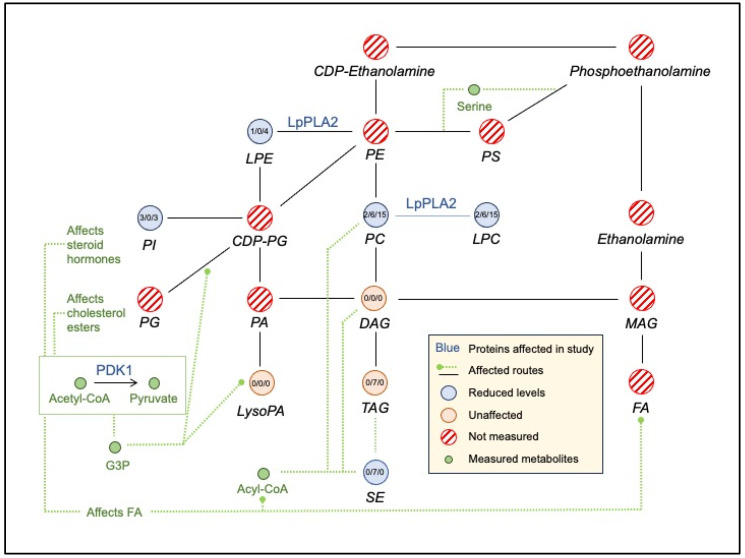
Schematic representation of change in lipid levels between baseline, post-intervention, and after washout.

**Figure 4 nutrients-16-00188-f004:**
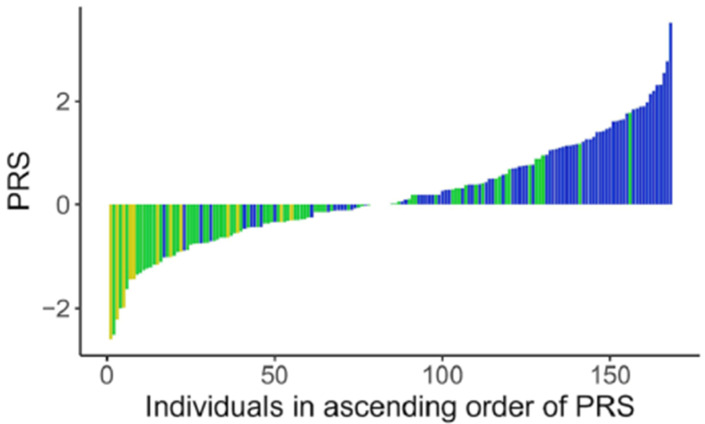
Distribution of PRS in participants. Polygenic risk scores in individuals (each bar) with their levels of vitamin B12 levels as low 
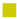
, normal 
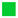
 and high 
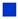
.

**Figure 5 nutrients-16-00188-f005:**
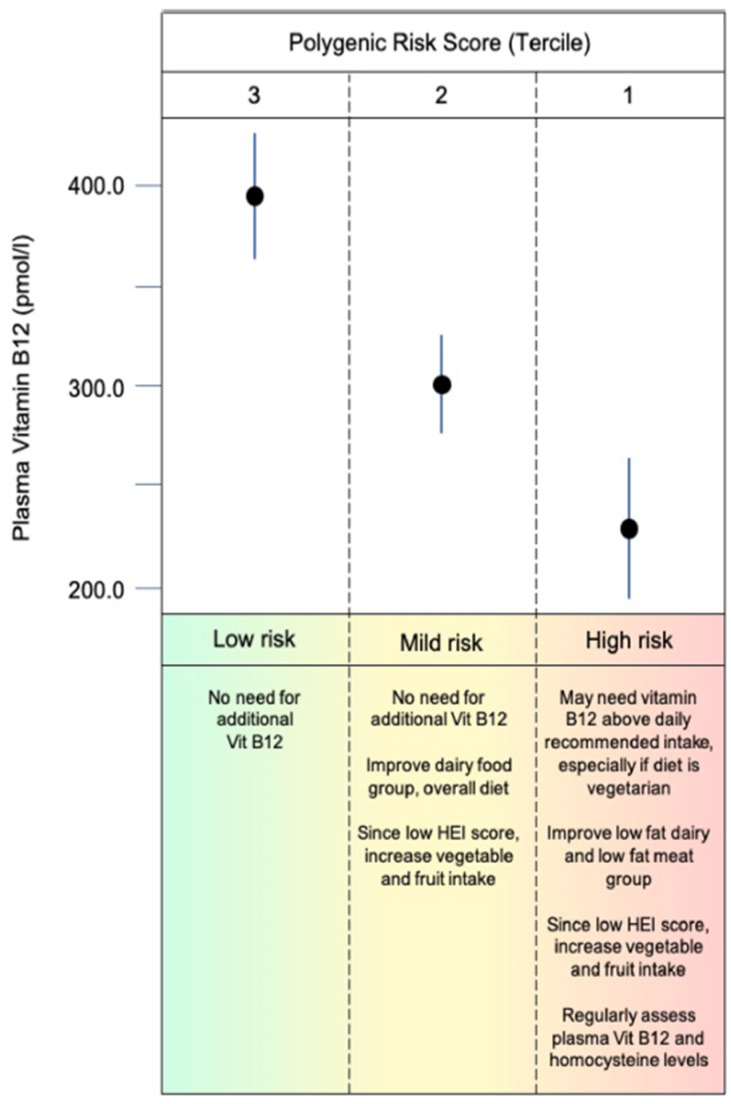
Hypothetical scenario for nutritional counseling for individuals with different baseline levels of vitamin B12 and accounting for B12 PRS, demographic information, and current dietary behavior.

**Table 1 nutrients-16-00188-t001:** Data Types of Micronutrient Genomics Project.

Variable	Number of Variables
Anthropometry	7
Body Mass Index (BMI)	1
24 h recall	28
Food Frequency Questionnaire	28
Physical Activity	20
Social Economic Status	22
Clinical Biochemistry	18
Hematology	13
Tanner Classification	1
Proteomics (plasma-Somalogic)	1129
Proteomics iTRAQ	20
Metabolomics (plasma-NMR)	24
Lipidomics	76
Fatty acids (Red Blood Cells)	25
Amino acids (Red Blood Cells)	25
Comet Assay (whole blood)	9
Vitamins (plasma)	36
Methionine Pathway (RBC)	13
Healthy Eating Index (HEI)	13
Whole genome genotyping	>4.3 million
Exome (percent of genome)	~2%
GIS Food Access & Quality	~50
Food price comparison	16

**Table 2 nutrients-16-00188-t002:** HEI food-intake and plasma biomarkers.

Dietary Group	Metabolite	R	*p* Value
Fruit intake	Linoleic acid	0.23	0.003
	α-linolenic acid	0.3	0.001
	EPA	0.26	0.001
	DHA	0.29	0.001
	β-carotene	0.19	0.020
Legumes/Vegetables	Linoleic acid	0.25	0.002
	α-linolenic acid	0.36	0.001
	EPA	0.27	0.001
	DHA	0.34	0.001
	β-carotene	0.25	0.002
All vegetables (including legumes)	Creatine	0.31	0.003
Dark greens and legumes	Creatine	0.37	0.001
Animal protein	Creatine	0.34	0.003
Milk/Dairy	Retinol	0.19	0.001
	Pyridoxal	0.21	0.007

The strongest correlations, after adjustments for age, sex, body-mass index, and for plasma cholesterol when needed. From [46].

**Table 3 nutrients-16-00188-t003:** FFQ food intake and plasma biomarkers.

Nutrient Intake	Metabolite	R	*p* Value ^1^
Animal protein	Creatine	0.19	<0.05
Myristic acid (C14:0)	C14:0	0.2	<0.01
EPA	EPA	0.15	<0.05
DHA	DHA	0.18	<0.05
β-carotene	β-carotene	0.31	<0.001
Folate	Folate	0.15	<0.05
Vitamin B3	Nudifloramide	0.17	<0.05
Vitamin B5	Pantothenic acid	0.17	<0.05
Vitamin B6	Pyridoxal 5′-phosphate	0.19	<0.05
**Food Groups and Biomarkers**			
Fish products	EPA	0.19	<0.01
	DHA	0.15	<0.01
Milk/Dairy	Myristic acid (C14:0)	0.20	<0.01
	Pyridoxal 5′-phosphate	0.32	<0.001
	Vitamin B12	0.23	<0.001
Total vegetables	β-carotene	0.36	<0.05
Dark green/orange	β-carotene	0.36	<0.05
Green vegetables	5-methyltetrahydrofolate	0.20	<0.05
Flour products	Para-aminobenoylglutamic acid	0.27	<0.01

^1^ Correlations between nutrients intake and their biomarkers (plasma and RBC metabolites), after adjustments for energy intake, age, sex, body-mass index, and cholesterol when needed. From [47].

**Table 4 nutrients-16-00188-t004:** Vitamins and one-carbon-pathway metabolites.

Metabolite 1	Metabolite 2	Δ Metabolite 1 ^1^	Δ Metabolite 2 ^2^
Vitamin B2	S-adenosylmethionine (SAM)	−1 nmol/L	−1.8 µmol/L
Vitamin B2	SAM:SAH ratio	−1 nmol/L	−0.20
Vitamin B6	Homocysteine (Hcy)	+1 nmol/L	−0.11 µmol/L
Vitamin B12	Homocysteine (Hcy)	+1 nmol/L	−0.14 µmol/L
Hcy	Linoleic acid (LA)	+1 nmol/L	+0.24 mg/dL
	α-linolenic acid (ALA)		+0.24 mg/dL
	Arachidonic acid (ARA)		+0.38 mg/dL
	Eicosapentaenoic acid (EPA)		+0.35 mg/dL
	Docosahexenoic Acid (DHA)		+0.49 mg/dL

^1,2^ + = increase (+) or decrease in metabolite 1 results (+) or decrease (−) in metabolite 2. From [66]. SAH = S-adenosylhomocysteine.

**Table 5 nutrients-16-00188-t005:** Vitamins and PUFA.

Vitamin	Fatty Acid	Δ B2 ^1^	Δ Fatty Acid ^2^
Vitamin B2	Linoleic Acid (LA)	+1 nmol/L	+0.15 mg/dL
	α-linolenic acid (ALA)		+0.15 mg/dL
	Eicosapentaenoic acid (EPA)		+0.19 mg/dL
	Arachidonic acid (ARA)		+0.20 mg/dL
	Docosahexenoic Acid (DHA)		+0.25 mg/dL
Folate	Linoleic Acid (LA)	+1 ng/mL	+0.15 mg/dL
	α-linolenic acid (ALA)		+0.15 mg/dL
	Eicosapentaenoic acid (EPA)		+0.14 mg/dL
	Arachidonic acid (ARA)		+0.22 mg/dL
	Docosahexenoic Acid (DHA)		+0.21 mg/dL

^1,2^ + = increase in vitamins results in increase in fatty acid. From [66].

**Table 6 nutrients-16-00188-t006:** B vitamin—fatty acid associations.

Vitamin	Fatty Acid	Association	ß-Coefficient	*p* Value
B2	Palmitoleic	Positive	0.12	<0.01
	Oleic	Positive	0.06	<0.01
	Elaidic trans-Fatty Acid	Negative	0.10	<0.01
B12	Palmitoleic	Positive	0.13	<0.01
	Oleic	Positive	0.06	<0.01
	Palmitic	Positive	0.04	<0.01
	Stearic	Positive	0.05	<0.01
	Eicosanoic	Positive	0.05	<0.01
B6	Linoleic Acid	Negative	0.07	0.02
	Alpha linoleic Acid	Negative	0.10	<0.01
	Arachidonic acid	Negative	0.1	0.02
	Docosahexenoic acid	Negative	0.12	0.03

From [66].

**Table 7 nutrients-16-00188-t007:** Lipid-profile-based metabotypes.

	Group 1 *n* = 10	Group 2 *n* = 10	*p* Value
Mean Triglycerides mg/dL	63.3 (39–103.7)	133.7 (75–220.3)	0.001
Mean LDL mg/dL	85.5 (66.7–124)	114.5 (48–152.3)	0.143
Mean VLDL mg/dL	12.7 (8.0–21)	26.7 (15.7–44)	0.001
α-tocopherol (Vit E) V2 (µg/mL)	5.6 (3.1–7.5)	8.0 (3.1–10.1)	0.063
**Genetic Ancestry**			
African (%)	16.8 (4.8–58.6)	35.1 (10.4–96.8)	0. 06
Europe (%)	71.3 (17–89.9)	29.1 (0–69)	0.004
Native America (%)	7.1 (0–23.7)	14 (3.1–43.8)	0.031

From [81].

**Table 8 nutrients-16-00188-t008:** Vitamin—inflammatory metabotypes.

	Group 1 *n* = 30	Group 2 *n* = 94	*p* Value
Riboflavin nmol/L	18.1 ± 10.4	13.1 ± 8.3	0.01
Pyridoxal nmol/L	8.9 ± 2.9	8.0 ± 3.0	0.04
Cobalamin pg/mL	783 ± 210	384 ± 192	0.01
Linoleic acid mg/dL	15.3 ± 4.7	16.6 ± 4.2	0.04
Homocysteine umol/L	2.5 ± 0.06	2.9 ± 0.8	0.04

From [82].

## Data Availability

The raw data from the Micronutrient Genomics Project are available from the corresponding author on reasonable request.

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
