# Peer review of "Human Nutrition Research in the Data Era: Results of 11 Reports on the Effects of a Multiple-Micronutrient-Intervention Study"

_nutrients, 2024, doi:10.3390/nu16020188_

Round 1

Reviewer 1 Report

Comments and Suggestions for Authors

Even though the idea of summarizing the results of this study/intervention program is a good idea, the authors should make profound changes to show what is important in the design used/results obtained (innovative, completing what is known until now, etc.). Underlining the extent to which it contributes to the existing literature, pointing out possible lines of research at the end of its intervention.

Since this manuscript is a summary of previously published articles, the title should be modified to reflect this fact.

The presentation of the information may be a little confusing for the reader not accustomed to this type of research, in which the aim is to individually evaluate a group of pre-pubertal children and in their early adolescence (not necessarily healthy) the level of intake, deficiency and response to a multi-supplement (vitamins and minerals), taking into account their genetics and metabolic response. Therefore, I suggest authors be sure to write the full name the first time an abbreviation appears.

The abstract should be modified to reflect what the authors are going to show in this article.

The introduction should be modified. The authors should show why they implemented this nutritional intervention program (reasons), rather than pointing out changes in the way it has been studied. What is important in this study? The main objective should be direct and the same throughout the manuscript (summary, introduction, results-discussion). Although this is a summary of already published articles, the introduction should be concise and concrete, providing the most important information on which the hypotheses/objectives of this study were based.

The materials and methods section is scarce and requires improvement. The description must be clear, concise and detailed. Who conducted this study? The authors must declare that this study was carried out following the recommendations of the Declaration of Helsinki. I want to clarify a doubt, was the collection of information and blood samples done the first year and repeated? How often? Were there one or two interventions with vitamin-mineral supplementation? It would be a good idea to indicate that in this section, a summary of the general design of the study (intervention program) will be made. The description of the subsection "Statistical analysis!" should be improved. What were the cut-off points used to evaluate significant results? What was the statistical program to evaluate the results? After reading the entire article, I ask what the primary objective and secondary objectives of this study-intervention program were. Specify them in this section, for example, Primary “to analyze the response to the consumption of multiple micronutrients.” Secondary: "predict individual responses to a nutritional change", etc. It would be better if the authors noted that the objectives, methodology (design), and discussion (analyzes) of general (section 3: Primary results) and specific (section 4: Thematic studies and specific results) results will be shown below.

General results: It would be a good idea to better characterize the population studied (age, sex, Tanner development, etc.). How many children and adolescents were there? I would recommend the authors review the next two sections. They seem like they were written by several people. Unify names and abbreviations. What is not clear is what happens with the minerals, only calcium is mentioned. It should be noted whether or not they had any effect on the intervention carried out, or if the results have not yet been published.

Specific results: Lines 205-6: Was this issue not considered in the design of this study? Line 214: Clarify this point. Nutritional biomarkers are evaluated to observe not only the deficiency of a nutrient (micro-macro nutrients) but also whether after its supplementation (improvement in diet) its levels/deposits improve. Line 218: The strength of the results was mild-moderate. This study did not demonstrate nutritional status, obviously, because it measured individual/group improvement. Why do not the authors show individual data? How much was the improvement? Lines 231-2: The characterization of the population should be done at the beginning, not in each subsection. Line 309: Were these changes significant? It would be a good idea to clarify if there is bias in the results shown on lines: 173 and 359. Line: 382: In each subsection, it should be clear with which references you analyzed your results. What does PRS mean? The first time that an abbreviation appears, the full name should be entered. In each subsection, the results should be discussed from multiple angles and placed in context without being overinterpreted. Authors should focus their discussion on the central topic of the study and avoid repeating the information provided.

Recapitulation instead of “Discussion”. Before the conclusion, a paragraph of limitations and suggestions for this study/program should be written.

Conclusion: Improve this section

Comments on the Quality of English Language

Minor editing of English language required.

Author Response

Thank you for your valuable comments. Please check the attachment.

Reviewer 2 Report

Comments and Suggestions for Authors

Abstract:

·         The abstract is quite detailed; however, it might benefit from being more concise. Consider condensing more detailed aspects to focus on the key results and their implications.

·         It briefly mentions the study's design and objective but could more explicitly state the primary hypothesis or research question driving the study. This would give readers a clearer understanding of the study's purpose from the outset.

·         The abstract outlines the unique aspects of the study design, including the n-of-1 approach and the inclusion of a replication arm. However, a brief mention of the study population's demographic details (e.g., age range, health status) in the abstract would provide valuable context for understanding the scope and applicability of the findings.

·         Main Findings: The abstract mentions that key findings and consistent results are summarized in the article, but it does not provide a clear snapshot of these results in the abstract itself.

1. Introduction:

·         The introduction provides a historical overview of micronutrient research and highlights the shift from reductionist approaches to system-level understanding. This context sets a solid foundation for the manuscript's focus on a holistic approach to micronutrient research. However, the transition from historical context to the present research question could be made more explicit.

·         It effectively identifies a gap in current micronutrient research, particularly regarding the limitations of traditional approaches in understanding complex physiological effects. Enhancing this section with more specific examples of past research limitations and their implications could strengthen the argument for the necessity of the study.

·         While the introduction mentions the transition to big data strategies in nutrition research, it could more clearly articulate the current study's specific hypothesis or research objectives. A distinct statement outlining the primary research question or hypothesis would guide the reader's expectations for the study.

·         It touches on the study's relevance to current trends in nutritional science, like the move toward personalized nutrition and big data. Expanding on how this study contributes to these evolving areas could enhance the introduction's impact.

·         It briefly mentions the n-of-1 experimental design but does not delve into why this approach suits the study's objectives. Providing a rationale for the chosen methodology to address the research gap would be beneficial.

·         The introduction has technical terms and concepts. While this is appropriate for a specialized audience, ensuring that terms are clearly defined and explained upon first use can aid in readability and understanding, particularly for interdisciplinary readers.

·         The introduction flows from historical context to current research trends, and the study's approach is generally reasonable. However, ensuring a smooth and logical progression of ideas that culminates in a clear statement of the study's purpose would further enhance this section.

2. Rationale and Experimental Design

2.1. Rationale for the intervention.

·         The section explains the rationale behind using an n-of-1 experimental design without a control arm and its implementation in two successive years. However, it could benefit from a more detailed explanation of why this specific intervention (the choice of micronutrients and their dosages) was selected. Discussing the scientific basis for the chosen micronutrients and their expected impact on the study population would strengthen the rationale.

·         The mention of using community-based participatory research principles is a strong aspect, indicating inclusivity and relevance to the local community. Expanding on how community inputs influenced the study design would provide valuable insights into the intervention's contextual appropriateness.

·         While the intervention is briefly described, it lacks detailed information on the composition and dosage of the micronutrients used. A clear and detailed description of the intervention (e.g., specific vitamins and minerals included, their dosages, and the rationale for these choices) would enhance understanding and reproducibility.

·         The selection of the age group (9 to 13 years old) and the duration of the intervention (6 weeks, followed by a 6-week washout) is mentioned. Still, the reasoning behind these specific choices is not clearly articulated. Explaining why this age group and duration were chosen based on existing literature or preliminary findings could add depth to the rationale.

·         The section would benefit from directly connecting the rationale for the intervention to gaps or findings in existing research. How does this intervention address shortcomings or unanswered questions in micronutrient research?

·         Given that the study involves children, a brief mention of the ethical considerations and safeguards would be appropriate, especially in a section discussing the intervention's rationale.

·         The decision to not include a control arm is mentioned but not thoroughly justified. Expanding on why a control group was deemed unnecessary or impractical for the study's aims would strengthen this aspect of methodology.

·         The rationale should be clearly linked to the study's hypotheses or research questions. Clarifying how the intervention design is expected to test or explore these hypotheses would provide a stronger foundation for the study.

2.2. Statistical Analysis

·         While the complexity of these methods is appropriate for the n-of-1 design, the manuscript could benefit from a clearer explanation of why these specific statistical methods were chosen and how they are particularly suited to the study's data and objectives.

·         This section may be challenging for readers unfamiliar with advanced statistical techniques like Bayesian methods. A brief, non-technical explanation of these methods and their relevance to the study's data analysis would enhance the section's accessibility.

·         The manuscript mentions using "classical" statistical methods for analyzing group-level physiology changes. Providing more detail on these methods and how they complement the individual-level analysis would be beneficial. This would also help understand how the study bridges individual and population-level insights.

·         This section acknowledges the challenges of analyzing individual-level data but does not delve deeply into how these challenges were specifically addressed in the study. Elaborating on this could provide valuable insights into the study's methodological rigor.

·         The section discusses using a method that accounts for effect size above RTM but does not provide sufficient details on how this was implemented or its importance in the study context. Expanding on this would strengthen the reader's understanding of the analytical approach.

3.1. Population level results

·         The section effectively presents a range of population-level outcomes from the intervention, including changes in micronutrient levels and physiological measures. However, it could benefit from a clearer structure grouping related findings for better readability and understanding.

·         While the results are described, there is a need for more context. How do these findings compare with existing literature? Including a brief discussion of how these results align with or differ from previous studies would provide valuable context and help understand the findings' significance.

·         While the results are reported, there is a lack of detailed interpretation. What do these changes in physiological measures and micronutrient levels imply for the broader understanding of nutrition and health? More detailed interpretation and discussion of the implications of these findings would enhance the section's impact.

·         The manuscript mentions the variability in responses to the intervention but does not delve deeply into this aspect. It would be insightful to discuss possible reasons for this variability and its implications for nutrition research and personalized nutrition.

·         The results should be explicitly linked to the intervention design. How do the specific components of the intervention (e.g., types and doses of micronutrients) relate to the observed outcomes? This connection would help in understanding the efficacy of the intervention design.

·         If any subgroup analysis was conducted (e.g., based on age, sex, baseline nutritional status), this would be a valuable addition to the section. It would provide insights into how different groups within the population responded to the intervention.

·         Discussing how confounding factors were accounted for in the analysis and how they might have influenced the results would strengthen the credibility of the findings.

3.2. Population level analysis of clinical and omic measures post intervention

·         The section provides an overview of the changes observed in various metabolites post-intervention. However, the presentation of these results could be more detailed and clearer. Including specific data points, such as the percentage change or the exact values of increase/decrease for each metabolite, would enhance the reader's understanding.

·         While the section mentions that changes were statistically significant, including specific statistical values (e.g., p-values, effect sizes) would be beneficial. This would help readers assess the robustness and practical significance of the findings.

·         Comparing the post-intervention changes to baseline measures or expected norms in the population can provide context to the findings. Discussing how these changes compare to what is typically seen in similar populations would add depth to the analysis.

·         If there were any unexpected changes in clinical or omic measures, discussing why these might have occurred would be insightful. This could include hypothesizing about underlying biological mechanisms or considering other external factors.

·         The section would benefit from discussing the clinical relevance of the observed changes. How might these changes impact overall health or the management of specific health conditions? This would help in understanding the practical implications of the study.

·         Acknowledging any limitations in interpreting the data, such as potential confounding factors, the representativeness of the sample, or limitations in the measurement techniques, would provide a more balanced view of the results.

·         The section should explicitly connect these findings back to the specifics of the intervention (e.g., types and amounts of micronutrients used). This would help in evaluating the efficacy of the intervention design.

·         If the study included diverse participants, analyzing whether there were different responses to the intervention across various subgroups (e.g., based on age, sex, and baseline health status) would be valuable. This could provide insights into personalized nutrition and how individuals might respond to similar interventions.

3.3. Interindividual variability.

·         The section addresses individual differences in response to the micronutrient intervention. However, it could benefit from a deeper analysis. Exploring and discussing potential factors contributing to this variability, such as genetic differences, environmental factors, or baseline nutritional status, would add depth to the analysis.

·         While the section acknowledges the variability in response, it would be strengthened by a detailed description of the statistical methods used to analyze this variability. Clarifying how interindividual differences were quantified and interpreted would enhance the readers' understanding of the methodological rigor.

·         The observation of interindividual variability aligns well with the growing interest in personalized nutrition. Discussing how these findings contribute to the field of personalized nutrition, including implications for dietary recommendations and future research directions, would enhance the relevance of this section.

·         Elaborating on the clinical and practical implications of the observed interindividual variability would be valuable. For instance, how might these findings inform clinical practice or the development of targeted nutritional interventions?

·         Comparing the observed variability in this study with findings from other studies in the field could provide context and contribute to a broader understanding of the variability in micronutrient responses.

3.4 . Predicting responses to the intervention.

·         The section addresses elastic net regression (Enet) to predict individual responses to nutritional changes. However, it would be beneficial to provide a clearer explanation of the Enet method for readers who may not be familiar with this statistical technique. A brief description of how Enet works and why it is suitable for this analysis would enhance understanding.

·         While the section discusses the variables predictive of response for most metabolites, more detailed information on these variables and their selection would be valuable. Additionally, presenting specific results or patterns observed in the predictive analysis, possibly in a summarized tabular or graphical form, would aid comprehension.

·         The manuscript mentions the prediction of responses based on baseline values and other variables, but there is little discussion on the accuracy and reliability of these predictions. Information on the predictive accuracy and any predictive model validation would significantly strengthen this section.

·         Comparing the Enet model's performance to these could add depth to the analysis if other predictive models were considered or used in similar studies. This would help in understanding the relative strengths and weaknesses of the chosen method.

4. Topic Specific Studies and Results

4.1. Food Intake Studies.

·         The section discusses the assessment of dietary status and the validation of food intake methods using biomarkers. A more detailed explanation of the study design, including the selection of biomarkers and their relevance to the dietary components being measured, would enhance the readers' understanding. Clarifying the choice of methods for dietary assessment (e.g., FFQ, 24h-recall) and their validity in the study context would be beneficial.

·         While the section provides an overview of the findings, it could benefit from a more detailed presentation of the results. Including specific data, such as correlation coefficients between dietary intake estimates and biomarkers, would provide a clearer picture of the study's findings. Tables or graphs illustrating these correlations could aid in visualizing the data.

·         The section mentions the limitations of using biomarkers in cohort studies. Expanding on these limitations, particularly in the context of this study's design and objectives, would offer a more comprehensive view of the challenges and implications of using biomarkers for dietary assessment.

4.2. Impact on lipid species.

·         The section discusses the impact of micronutrient intervention on various lipid species, yet the specifics of these lipid species are not adequately detailed. A clearer enumeration and description of the lipid species analyzed and the rationale for their selection would enhance the reader's understanding.

·         While the section mentions using DNA aptamer-based methods and untargeted lipidomics, it lacks detailed information about these methodologies. Expanding on the technical aspects of these methods, including their accuracy and reliability, would provide a more comprehensive understanding of the analysis performed.

·         The section presents the outcomes of the intervention on lipid species but falls short in interpreting these findings in a broader context. Discussing the potential implications of these changes in lipid levels for overall health, disease risk, or metabolism would add depth to the analysis.

·         Providing a comparative analysis with existing research on micronutrients and lipid metabolism would contextualize the findings. How do these results align with or differ from current understanding in the field?

4.3. Changes in 1-carbon pathway metabolites [51].

·         The section refers to the analysis of 1-carbon pathway metabolites, but it could benefit from a more detailed explanation of the specific methodologies used for this analysis. Clarifying the techniques used for LC/MS analysis and how these were applied to study 1-carbon pathway metabolites would enhance the reader's understanding.

·         While the section mentions several metabolites, including vitamins B2, B12, B6, folate, and homocysteine, a more detailed explanation of why these particular metabolites were chosen and their relevance to the study's objectives would be beneficial. Additionally, providing a brief background on the role of these metabolites in the 1-carbon pathway would aid in contextualizing the findings.

·         The results are briefly mentioned; however, the section would benefit from a more detailed presentation of these findings, including specific data on how the intervention influenced the levels of these metabolites. Graphical representation, such as charts or tables, would help visualize these changes. Moreover, interpreting these changes in the context of their potential impact on metabolic pathways and health outcomes would add depth to the analysis.

·         Providing a comparative analysis with existing literature on micronutrients' effects on 1-carbon metabolism would contextualize the findings. How do these results align with or differ from current understanding in the field?

·         Discussing the potential implications of these findings for nutritional recommendations, disease prevention, or management, especially in the context of metabolic diseases or conditions associated with 1-carbon metabolism, would enhance the section's relevance.

4.4. PUFA associations with DNA damage, folate, and vitamin B2 [56–58].

·         The section explores the relationships between polyunsaturated fatty acids (PUFAs), DNA damage, and specific vitamins. It would benefit from a clearer introduction explaining why these particular relationships are being investigated. A brief background on the known or hypothesized links between PUFAs, DNA damage, and these vitamins would provide valuable context.

·         The section should detail the methods used to assess DNA damage and measure PUFA and vitamin levels. Clarifying whether these methods are well-established or novel and discussing their accuracy and reliability would lend credibility to the findings.

·         The results are briefly mentioned, but a more detailed presentation would be beneficial. Including specific data points, such as the levels of DNA damage and concentrations of PUFAs and vitamins, along with their statistical correlations, would give readers a clearer understanding of the findings. Tables or graphs could effectively illustrate these relationships.

·         While the results are reported, a deeper interpretation is needed. How do these findings contribute to our understanding of the roles of PUFAs and vitamins in DNA damage and repair mechanisms? Discussing the potential biological implications and the pathways involved would add depth to the analysis.

·         Comparing the findings with existing research on the associations between PUFAs, DNA damage, and vitamins would contextualize the results. Discussing how these findings align with or differ from current scientific understanding would enhance the section's relevance.

·         The section would benefit from a discussion of the practical implications of these findings. For example, how might these results inform nutritional recommendations or strategies for preventing or managing conditions associated with DNA damage?

4.5. Metabo Groups.

·         The concept of metabotyping is central to this section. It would be beneficial to start with a clear, concise explanation of what metabotyping entails, its significance in nutritional research, and how it was applied in this study. A brief overview of the rationale behind using metabotyping in this context would set a solid foundation for the section.

·         The section should provide more detailed information on how metabotypes were identified and analyzed. This includes specifics on the analytical techniques, criteria for metabotype classification, and how these were applied to the study data. Clarifying these methodological aspects would enhance the reader's understanding of the study's approach and the validity of its conclusions.

·         The results related to metabotyping are briefly mentioned, but a more comprehensive presentation would be beneficial. This includes detailed findings regarding how different metabotypes responded to the micronutrient intervention. Additionally, interpreting these results in the context of their potential biological significance and implications for personalized nutrition would add depth to the analysis.

·         Comparing the metabotype findings from this study with existing research would provide valuable context. How do these results align with current understanding of metabolic responses to dietary interventions? Discussing similarities or differences with previous studies would enhance the section's relevance.

4.6. Identification of Vitamin B12 Genetic Risk Score [84].

·         The section discusses the development of a genetic risk score (GRS) for vitamin B12, which is a complex concept. It would benefit from a more detailed and clear explanation of what a GRS is, how it is constructed, and specifically, how it was developed for vitamin B12 in this study. Clarifying the selection of single nucleotide polymorphisms (SNPs) and weighting these SNPs in the GRS would aid understanding.

·         The manuscript should provide a clear rationale for why vitamin B12 was chosen for this analysis. Including a brief overview of the importance of vitamin B12 in human health and its genetic determinants would set a strong foundation for this section.

·         While the section briefly mentions the statistical methods used, a more in-depth explanation of the analysis, including how the SNPs were associated with vitamin B12 levels and how ancestry was accounted for, would be beneficial. Interpreting these statistical results, including the implications of finding a significant genetic component to vitamin B12 levels, would add depth to the analysis.

5. Discussion

5.1. Replication.

·         This subsection discusses the importance of replication in research. However, it would benefit from a more detailed analysis of how the findings from the replication arm compared to the original results. Discussing specific instances where replication affirmed or differed from the original findings would add depth to the analysis.

·         Addressing the methodological consistency between the original study and the replication arm and discussing any challenges or variations encountered would provide insight into the robustness of the findings.

5.2. Prediction of Individual Responses to Intervention.

·         The discussion should provide more details on the performance of the predictive models, including their accuracy, reliability, and any limitations observed in their predictive capability.

·         Expanding on how these predictive models can inform personalized nutrition practices would be valuable. Discussing the potential application of these models in clinical or public health settings would enhance the relevance of this section.

5.3. Translational Research

·         This subsection should elaborate on how the findings have been or could be translated into practice, especially considering the community-based participatory research approach. Discussing the impact of the research on the community involved and potential broader applications would be insightful.

·         Exploring the implications of the study's findings for public health policy, particularly in the context of nutrition and diet-related diseases, would add significance to the discussion.

6. Conclusion

·         The conclusion should briefly summarize the study's key findings, highlighting the most significant outcomes and their implications for nutritional science and practice.

Comments on the Quality of English Language

The manuscript is generally understandable, but there are areas where clarity and precision could be improved. This includes refining complex sentences for better readability, ensuring consistency in terminology, and possibly rephrasing certain sections for clearer communication of the research findings and methodologies. Such edits would enhance the overall readability and effectiveness of the manuscript in conveying its scientific message.

Author Response

(The authors gave the same response as above.)

Round 2

Reviewer 1 Report

Comments and Suggestions for Authors

Even though the manuscript has been considerably improved by explaining the importance of previously published results, there are some points to change.

Specify in the material and methods section: Primary “to analyze the response to the consumption of multiple micronutrients.” Secondary: "predict individual responses to a nutritional change", etc. Who conducted this study?

Line: 153: from 9 to 13 years. Lines 171-172: It would be a good idea to write the percentage in parentheses. Lines 173-174: Specify what micronutrients insufficiencies/deficiencies were found in these populations. Line 185: What happened to Zn? Lines 200-212: What are the authors trying to convey? Which is the message? Line 267: Why did the authors remove this information “The majority of the participants (91%) had a very poor dietary pattern according to the Brazilian Healthy Eating Index, and specifically, poor in vegetables, legumes, fruits, whole grains, milk and dairy, and rich in sugar and saturated fat.? The food pattern intake did not change during the 12-week in either year.” Line 282 and 398: It would be better to write the full name of FFQ and PUFAs. Lines 418-427: It would be better to write these results in a table. Add references from the "Relevance" subsections. Line 601-602: Improve the title of figure 5.

Comments on the Quality of English Language

Minor editing of English language required.

Author Response

Even though the manuscript has been considerably improved by explaining the importance of previously published results, there are some points to change.

Specify in the material and methods section: Primary “to analyze the response to the consumption of multiple micronutrients.” Secondary: "predict individual responses to a nutritional change", etc. Who conducted this study?

This sentence was clarified:

While the primary end points of this study were to analyze the response of individuals who consumed supplemental multiple micronutrients for 6 weeks, results of analysis of baseline anthropometric, clinical, and omic data defined the nutritional status of the population.

Line: 153: from 9 to 13 years.

Corrected

Lines 171-172: It would be a good idea to write the percentage in parentheses.

Lines 173-174: Specify what micronutrients insufficiencies/deficiencies were found in these populations. 

Added

Line 185: What happened to Zn?

Zinc was not measured.

Lines 200-212: What are the authors trying to convey? Which is the message?

With all due respect, this question is not clear.  This passage describes the many metabolites that were changed (with statistical significance) by the intervention.  The fact that the concentration of some plasma metabolites were not changed suggested to us that the intervention targeted processes with the specificity expected by cofactor – protein/enzyme interactions. We would have been very skeptical of the results if all metabolites changed which would have indicated a flaw in design or execution.

Line 267: Why did the authors remove this information “The majority of the participants (91%) had a very poor dietary pattern according to the Brazilian Healthy Eating Index, and specifically, poor in vegetables, legumes, fruits, whole grains, milk and dairy, and rich in sugar and saturated fat.? The food pattern intake did not change during the 12-week in either year.”

This sentence was removed because it was redundant with the statements in lines 164 to 172.

Line 282 and 398: It would be better to write the full name of FFQ and PUFAs.

FFQ is now defined in  line 188 (first appearance)

PUFAs now defined in line 423

Lines 418-427: It would be better to write these results in a table.

            Agreed – see table 7. 

Add references from the "Relevance" subsections.

This was done when appropriate for all relevance sections.

Line 601-602: Improve the title of figure 5.

Title now reads: Distribution of PRS in Participants.

Reviewer 2 Report

Comments and Suggestions for Authors

1. Abstract:

·         The abstract effectively introduces the study's purpose and key findings. However, it could benefit from more concise language and a clearer summary of the results and their implications.

2. Introduction:

·         Provides a solid background but could better connect historical context with the study's specific focus. It would be enhanced by more directly stating the study's objectives and relevance.

3. Rationale and General Experimental Design:

·         The experimental design is well-explained, but the rationale for choosing the specific age group and region could be elaborated further to strengthen the study's context.

4. Primary Results:

·         The presentation of primary results is thorough, but it could be made more reader-friendly with better visual representations of the data. Some interpretations of results seem to require more robust backing with references or data.

5. Secondary Results:

·         This section well-outlines the secondary results but could benefit from a more detailed discussion of how these results contribute to the field and relate to existing literature.

6. Main Results, Strengths, and Limitations:

·         The section accurately acknowledges the study's strengths and limitations. However, a deeper analysis of how these limitations might impact the overall conclusions would be beneficial.

7. Conclusion:

·         The conclusion effectively summarizes the study's findings. It could be strengthened by explicitly linking the results to potential practical applications or future research directions.

Comments on the Quality of English Language

The text is generally comprehensible but presents issues that could hinder readability and clarity, such as complex sentence structures and occasional grammatical inconsistencies. Moderate editing would significantly enhance its readability and professional presentation.

Author Response

  1. Abstract:
  • The abstract effectively introduces the study's purpose and key findings. However, it could benefit from more concise language and a clearer summary of the results and their implications.

The abstract was modified as requested

  1. Introduction:
  • Provides a solid background but could better connect historical context with the study's specific focus. It would be enhanced by more directly stating the study's objectives and relevance.

In The Structure of Scientific Revolutions [5], Kuhn described how “normal science” was interrupted by periods of “revolutionary science” resulting in new paradigms and ways of doing research. Notwithstanding the extensive cofactor/gene/pathway literature base, most nutritional intervention studies add one or a few micronutrients to an existing diet and analyze the population-level responses of specific biomarkers or pathways. While hypothesis – driven research has been a foundation of biomedical research, genomics, proteomic, metabolic, and other omic technologies are revolutionizing scientific studies. With some notable exceptions (e.g., [6,7]), nutrition research has been slow in adopting big data strategies and systems approach [8]. Our teams have a broad data-driven program micronutrients (Supplement 1 and Appendix A).

The specific goal of studies summarized in this report was to use diverse data sets to discover previously undetected physiological effects associated with a poor diet that included a more complete micronutrient composition. The nutritional intervention study in children and teens analyzed the effects of 12 micronutrients and 5 vitamins on physiological systems (Box 1).

  1. Rationale and General Experimental Design:
  • The experimental design is well-explained, but the rationale for choosing the specific age group and region could be elaborated further to strengthen the study's context.

This was explained in the primary and revised manuscript, but we also added a sentence to this section:

The selection of the age group was based on the high prevalence of overweight and obese consistent with trends in child and general Brazilian population and on the poor quality of their diets ([9] - accessed on 15 November 2017)]. In addition, this age group is under-represented in nutrition studies.

  1. Primary Results:
  • The presentation of primary results is thorough, but it could be made more reader-friendly with better visual representations of the data. Some interpretations of results seem to require more robust backing with references or data.

We added references from research related to our findings in the relevance section of each study.

  1. Secondary Results:
  • This section well-outlines the secondary results but could benefit from a more detailed discussion of how these results contribute to the field and relate to existing literature.

We added references from research related to our findings in the relevance section of each study.

  1. Main Results, Strengths, and Limitations:
  • The section accurately acknowledges the study's strengths and limitations. However, a deeper analysis of how these limitations might impact the overall conclusions would be beneficial.

We added additional statements to this section to put the results in context of other research.

  1. Conclusion:
  • The conclusion effectively summarizes the study's findings. It could be strengthened by explicitly linking the results to potential practical applications or future research directions.

We added additional statements to this section to put the results in context of other research

Comments on the Quality of English Language

  • The text is generally comprehensible but presents issues that could hinder readability and clarity, such as complex sentence structures and occasional grammatical inconsistencies. Moderate editing would significantly enhance its readability and professional presentation.

We clarified when we detected inconsistencies and readability. In some cases, readability may be defined by personal preferences.
